# Motion Correction in low SNR MRI using an approximate Rician log-likelihood

Anonymous[1], Anonymous[2], and Anonymous[2]

[1] Anonymous
[2] Anonymous
Anonymous

**Abstract.** Some MRI acquisitions, such as Sodium imaging, produce data with very low signal-to-noise ratio (SNR) and meaningful analysis may require several images to be averaged. As the data contains substantial noise, motion correction using standard registration tools may not be effective. This paper employs a simple generative model for the data, where the error is described as following a Rician distribution, which more accurately characterised the image noise. Maximum a posteriori inference is enabled by a differentiable approximation to the Rician log-likelihood function. We find that this approach substantially outperforms a Gaussian log-likelihood baseline on synthetic data that has been corrupted by Rician noise of varying degrees. We show results of our approach on real Sodium MRI data, and demonstrate that we can reduces the effects of substantial motion.

**Keywords:** Motion correction · Rician distribution · Low SNR

## 1 Introduction

Motion correction is an essential pre-processing step in an analysis pipeline to enable meaningful measurement of voxelwise properties. Existing linear registration tools, e.g. [4][8][2] are generally very effective for images with reasonable signal to noise ratio (SNR). However, relatively little research has been conducted into the registration of extremely noisy MRI data, such as acquired from Sodium MRI sequences. Most structural MR images have high SNR, in which case the acquisition noise is well described as following a Gaussian distribution. However, in situations where the SNR is very low it actually follows a Rician distribution [3]. This distinction can become significant for for image registration and motion correction; the Rician distribution is not symmetric and cost function derived from a Gaussian distribution (such as sum-of-squared differences) may fare poorly.

This paper introduces a linear motion correction algorithm, using a simple generative model of the data, which is heavily inspired by the seminal "Unified Segmentation" paper [1]. A diagram of our approach is given in fig 1. In this work, we parameterise our model using the means of different tissue classes and rigidly register this template to each observed image. A Rician likelihood drives

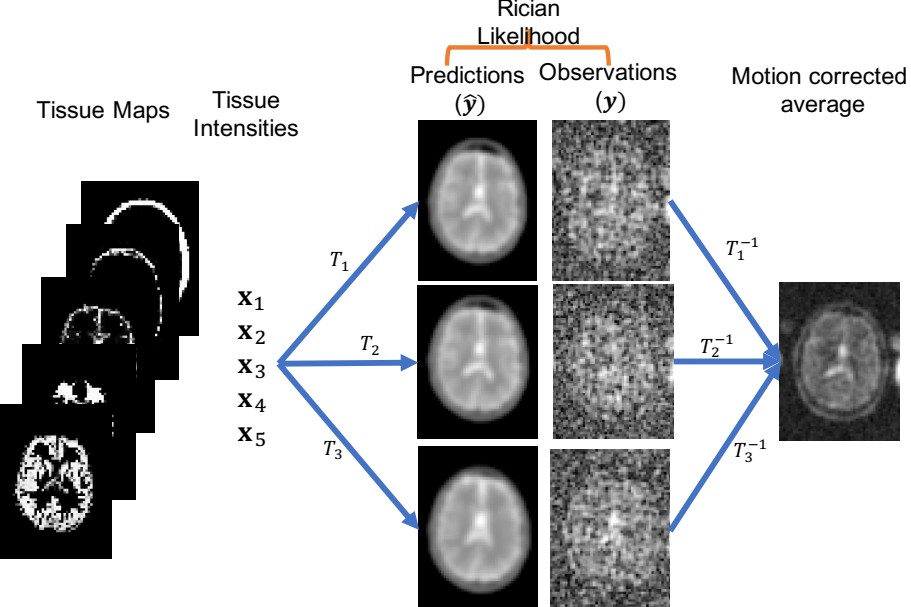

**Fig. 1.** Illustration of the model. A simple generative model, parameterised by mean tissue intensities and pre-calculated tissue maps provides predictions of the noisy observation data. The error in model prediction is described using a Rician likelihood, which is used to drive the transformation estimation. This allows us to arrive at a sharp motion corrected average image.

the alignment between the model and the data. However, as this distribution does not have a form that enables closed form updates we employ an approximation to the log-probability, and rely on automatic differentiation to provide gradients for optimisation [9].

We apply this approach to motion correct Sodium MRI data, which is an emerging imaging modality with many potential biomedical applications [6][11]. One limiting factor with such acquisitions is that they tend to have very poor signal-to-noise ratio, due to the relatively low concentration and magnetic susceptibility of Sodium. To compensate for this issue, multiple image acquisitions may be collected and subsequently averaged to provide improved SNR. In this context, subject motion can be very problematic as each individual image contains little useful information to drive alignment.

The contribution of this work is a generative inference framework for motion correction of Sodium (or other very noisy) MRI using a differentiable approximation of the Rician log likelihood. To the best of our knowledge, a Rician likelihood has not previously been used for image registration although recent works have shown it's benefits for regression[12]. We demonstrate that this ap-

proach is effective in removing substantial motion high noise situations, while aligning images such that the average is co-registered with the segmentation.

## 2   Background

### 2.1   Rician distribution

The noise distribution of low SNR MR images is known to follow a Rice distribution [3]:

$$p(\mathbf{y}|\hat{\mathbf{y}}, \sigma) = \mathrm{Rice}(\mathbf{y}; \hat{\mathbf{y}}, \sigma) = \frac{\mathbf{y}}{\sigma^2} \exp\left(\frac{-(\mathbf{y}^2 + \hat{\mathbf{y}}^2)}{2\sigma^2}\right) I_0\left(\frac{\mathbf{y}\hat{\mathbf{y}}}{\sigma^2}\right) \tag{1}$$

where $I_0$ is a modified Bessel function of the first kind with order zero (described in section 3.2). Unlike the Gaussian, this distribution is not symmetric with respect to it's first parameter, $\hat{\mathbf{y}}$. It also does not fulfill any of the conjugacy properties that would allow straightforward application in inference methodologies. It also does not provide an obvious cost function for comparing two images, as it requires a parameterisation in terms of the clean signal, $\hat{\mathbf{y}}$. Generative models can be used to provide such a parameterisation, and have been successfully used for image registration for many years [1].

## 3   Method

We consider a generative model for the image data based on tissue maps. The basis for this model is a set of 5 probabilistic tissue segmentation map, $G$ derived from the high resolution T1 image, which are co-registered with the data being motion corrected. We denote $G$ as a matrix of size $N \times 5$ containing tissue probabilities, where $N$ corresponds to the number of voxels and 5 the number of tissue classes. The intensity of any voxel can be predicted by matrix multiplication with $\mathbf{x}$, a vector containing the intensity for each tissue class.

As we are considering a motion correction problem, we are optimising for geometric transformations. Accordingly, we predict the data through the following model:

$$\hat{\mathbf{y}} = \mathrm{P}(\mathrm{T}(G\mathbf{x}, \mathbf{t}, \theta)) \tag{2}$$

where T corresponds to a rigid transformation, with translation $\mathbf{t}$ and rotation parameters given by $\theta$. P corresponds to the point-spread function of the acquisition sequence, which is empirically estimated a-priori.

These predictions $\hat{\mathbf{y}}$ can now be fit to the data using an appropriate log-likelihood function, such as the Rician distribution or the Normal distribution.

### 3.1   Priors

In this problem, we are considering the registration of noisy data. Accordingly, the model requires the specification of prior knowledge to enable robust inference.

We choose a Gaussian prior over the tissue intensities

$$p(\mathbf{x}) = \mathcal{N}([40, 30, 80, 50, 50], [4, 4, 5, 10, 10]^2) \qquad (3)$$

The means were derived using rounded average values for a given tissue class from the dataset and variances were chosen empirically.

The translations have a Normal prior, with a fixed variance specified in mm. The rotations, which are described through an axis-angle representation, also employ a Normal distribution:

$$p(\mathbf{t}_i) = \mathcal{N}(0, 2.5^2)$$
$$p(\theta_i) = \mathcal{N}(0, 0.05^2)$$

### 3.2   Approximating the Rician Log-Likelihood

Most of the Rician likelihood (eq 1) are amenable to efficient calculation in a differentiable manner. However, there is an infinite series, which requires approximating: $I_0$, which corresponds to a modified Bessel function of the first kind with order zero [13]. $I_0$ can be written as:

$$I_0(z) = \sum_{k=0}^{\infty} \frac{(\frac{1}{4}z^2)^k}{(k!)^2} \qquad (4)$$

We can approximate this series as a sum of the first $N_k$ terms. However, we still require a differentiable form for the factorial in the denominator. By noting both that $k! = \Gamma(k + 1)$, where $\Gamma$ is the Gamma function, and that we only require the log probability, we can write an approximation for $\log I_0(z)$ as:

$$\log I_0(z) \approx \text{logsumexp}(\mathbf{k}(\log(0.25) + 2 * \log(z)) - 2\ln\Gamma(\mathbf{k} + 1)) \qquad (5)$$

where logsumexp is a popular trick for calculating the logarithm of sum of exponentiated terms in a numerically stable fashion and $\ln\Gamma$ refers to the log Gamma function. $\mathbf{k}$ is a vector containing values from 0 to $N_k$, which is summed over. This implementation is empirically robust, and relatively computationally efficient. It is however inefficient in terms of memory, as we require multiplying each voxel by $N_k$ values. We find that $N_k = 50$ provided sufficient precision for our data.

### 3.3   Inference

We perform maximum-a-posteriori (MAP) inference on the model parameters $\Theta = \{\mathbf{x}, \mathbf{t}, \theta, \sigma\}$, with the following cost function:

$$\mathcal{L} = -\sum_{i}^{N} [\log p(\mathbf{y}_i|\mathbf{x}, \mathbf{t}_i, \theta_i, \sigma) + \log p(\mathbf{t}_i) + \log p(\theta_i)] + \log p(\mathbf{x}) \qquad (6)$$

Updates alternated between two groups of parameters, those that are shared for all images $\Theta_1 = \{\mathbf{x}, \sigma\}$ and those that vary per image $\Theta_2 = \{\mathbf{t}, \theta\}$. The updates for $\Theta_1$ were calculated using batches of 5 images at a time, and $\Theta_2$ were updated per image. To account for the batching in updating $\Theta_1$, we perform two update steps on these parameters for every step for $\Theta_2$. The Adam [5] optimiser was used to optimise the model parameters, and a fixed learning rate of $1e{-}2$ was found to be effective and convergence occurred withing 60 rounds of iterations.

## 4    Experiments

### 4.1    Synthetic Data

We generate synthetic data from our generative model with known tissue parameters, with voxelwise variability characterised by the prior distributions. These were then transformed to simulate random motion, with translations sampled from $\mathcal{N}(0, 5mm^2)$ and angles from $\mathcal{N}(0, 0.1^2)$. Each of these images was then corrupted with Rician noise at various levels. We then try to correct for the simulated motion using our model with either a Gaussian or Rician likelihood.

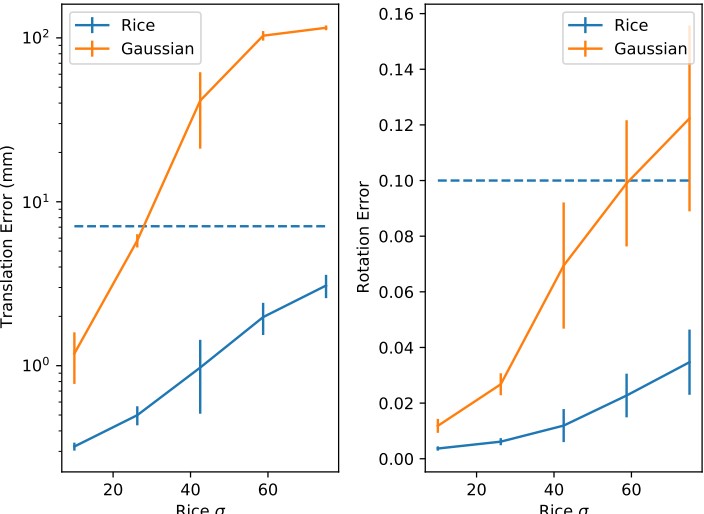

**Fig. 2.** Synthetic data experiments where the ground truth translation (mean euclidean distance) and rotation error (mean Frobenius norm of the difference of log matrices) are given in the above plots for varying Rician noise level. The dashed line indicates the average initial error. As can be seen, the error when using a Gaussian likelihood rises very quickly, whereas the Rician likelihood is more robust against high noise levels. $\sigma = 40$ is roughly equivalent to the Sodium MRI data.

Figure 2 illustrates that using the correct likelihood model has a substantial impact on registration performance, particularly in scenarios where high levels of noise are evident.

## 4.2   Sodium MRI

[23]NA MR images were acquired using a dual-tuned, 2-channel (one channel for sodium and one for proton) birdcage $^{23}$Na $^{1}$H coil developed by RAPID Biomedical GmbH (Rimpar, Germany). Sodium images were acquired using the FLO-RET sequence [10] with a resolution of 4mm$^3$. The k-space data were transferred offline and image reconstruction was performed in Matlab (MathWorks, Natick, MA, USA) using 3D re-gridding [10] with density compensation [14].

A low-resolution T1-weighted image (4mm$^3$), matching the resolution of the Na image was acquired using the dual-tuned sodium coil as well as a higher resolution T1-weighted image (2mm$^3$). These images were used for preparing tissue segmentation maps using SPM12 and co-registration with the uncorrected average Sodium image.

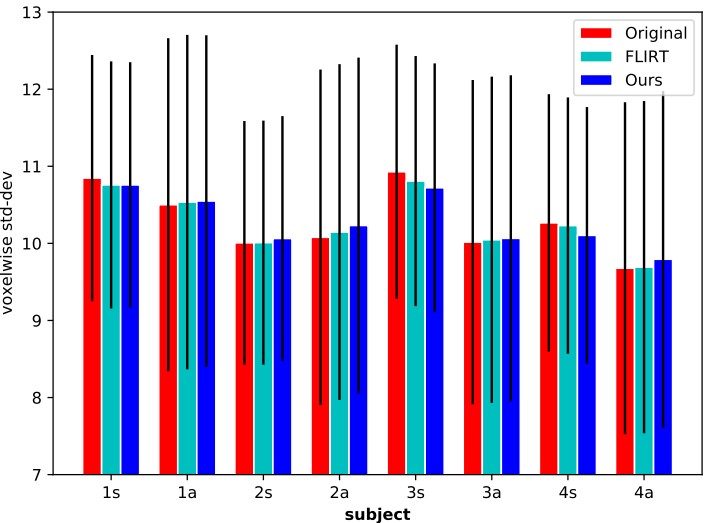

**Fig. 3.** Bar chart showing the mean voxelwise standard deviation over the set of acquired images for 4 subjects either sleeping (s) or awake (a). We compare our approach to the original uncorrected data and FLIRT. We find for sequences corrupted by motion (higher original std-dev) our approach tends to work well.

In the acquired data, we examine 4 subjects with images taken either when they are asleep (32 images) or awake (16 images). The data acquired when sleeping is much more likely to contain motion artefacts both due to the length of

| Original | mcflirt | Ours |
|----------|---------|------|

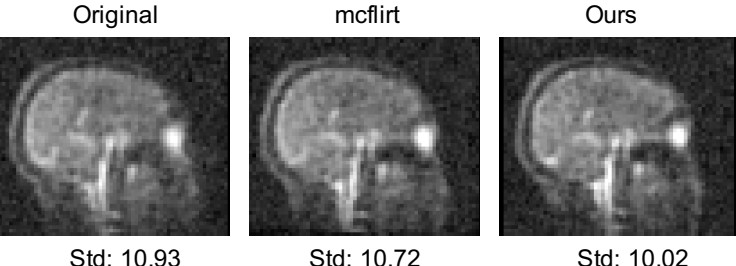

| Std: 10.93 | Std: 10.72 | Std: 10.02 |
|------------|------------|------------|

**Fig. 4.** Example average images calculated by averaging 32 Sodium MRI acquisition. Std refers to the mean voxelwise standard deviation at each voxel within the brain region. In this example, where a lot of motion was detected, our approach leads to a sharper average image.

the scan and unintentional movements during sleep. We consider motion correcting the individual sodium magnitude images using either "mcflirt"[4] or our approach.

We provide some quantification of the motion correction efficacy in fig 3. Here, we can see that in some sleeping subjects (1, 3 and 4) the original image data contains a high degree of voxel variation induced by apparent subject motion. Our approach successfully reduces the variation, leading to sharper average images, see fig 4 for the results on subject 3. FLIRT generally performs well for data with lower motion, whereas our approach makes some low motion examples slightly worse (subjects 2 and 4 when awake). Although the estimated transformation is only very small for those cases, with average translation 1.7mm (0.4mm) and average rotation norm 0.03 (1e-3), compared to an estimated translation of 7.75mm (6.49mm) and rotation norm of 0.145 (0.12) for subject 3's sleeping data.

## 5    Discussion

This work has not investigated the use of preprocessing the data using denoising methods [7]; although such approaches might produce cleaner representations for aligning the data, they also manipulate the underlying image statistics being modelled, which may lead to biased results.

We found that in some cases where low motion was observed, our algorithm over estimated the level of movement. This would likely be improved by specifying transformation priors that varied based on the expected level of motion, e.g. a tighter angle prior on the awake subjects.

We employed nearest neighbour interpolation to calculate the average images. This was required to avoid drastically changing the image statistics through linear interpolation. Future work will investigate using higher order interpolants with other methods, which are already available in FLIRT.

## 6   Conclusions

This paper has introduced a unified model for data modelling and co-registration of noisy MRI data with a Rician likelihood. We have shown the importance of choosing the right log-likelihood for generative models for motion correction, as the Gaussian distribution performs very poorly where the errors take a different form.

Our results provide qualitative support on real Sodium data, and show that we can resolve some substantial motion artefacts. Although our restricted parameterisation has some benefits in reducing overfitting, it also does not explain the data particularly well, which may be a further cause of estimating motion where none is present. Future work will investigate using more flexible model parameterisations, such as voxelwise/mixture distributions, which may better describe the image data and permit more accurate alignment.

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
