# OpenReview forum: "Motion Correction in low SNR MRI using an approximate Rician log-likelihood"
_WBIR.info/2022/Workshop/Biomedical_Imaging_Registration — WBIR 2022_

### Official Review · Reviewer_8tMb · 2022-02-20

**Rating:** 1
**Confidence:** 4

**Deanonymize Review:**

no

**Detailed Comments:**

General: Language check please.

- Abstract:

 “which more accurately characterised”, more than what? also change to past tense mid sentence.
Last sentence of abstract is unreadable.

- Intro:

“Existing linear registration tools, e.g. [4][8][2]”, something more modern than 2002?
“for for”
Fig 1 is very good. Suggest “Tissue maps” -> “Tissue segmentation maps”, also include the symbols (x, T, G) in the caption.
The word “in” is missing in the last sentence of Intro.

- Background:

Describe (to a reasonable extent) existing methods, including the basis of your work (originating from [1]), including your comparison method FLIRT).
In 2.1, write what y and y-hat are in connection to (1).
What are the “conjugacy properties”?
“it’s” -> “its”

- Method:

Explain first the problem formulation and the assumed data, including the T1 image. Refer once more to Fig 1.
2nd sentence: “map” -> “maps”
How is the co-registration performed?
Consider replacing 5 with a parameter $c$ for generality.
“fit to the data”, write out exactly what is fit to what, use introduced symbols.
Explicate that rotation is given in radians (I assume).
“logsumexp” is not really “a popular trick”; write the equation using correct mathematical notation, and then give implementation information, including references, and preferably share code.
Lacking info on Beta1, Beta2, eps, (and possibly more) settings of ADAM.
Say something about computational time.

- Experiments:

Give more information about the generated data. Why not share it openly? That would really improve usefulness (and citability) of your work.
Consider some synthetic image examples.
How is the Na MRI data segmented? Lots of info lacking. Also, if you can (are allowed to) share data, please do so.
Fig 3. Motivate your performance measure (and why not optimize that one directly?); relate to the performance measure in Fig 2.
Include also the same real data performance measure for your synthetic data, to complement the (known correct) registration performance; perhaps this may reveal why we do not see much improvement on the real data experiment.
Fig 4. What is “sharper”? How is that quantified?

- Discussion:

Over estimated -> Overestimated



Despite being an interesting and presumably novel method, due to the several weaknesses mentioned, I feel that I cannot recommend this manuscript for the WBIR workshop.

I believe the approach may still deserve to reach a larger audience. Please do consider improving the manuscript based on suggestions given and resubmit elsewhere, including sharing code and data.




**Paper Type:**

methodological development

**Strengths Weaknesses:**

Summary:

The authors propose a MAP model which takes into consideration the Rician noise distribution of MRI, and evaluates the model for motion correction of low SNR sodium imaging.
Using a high resolution T1 image, a tissue segmentation map is generated. This, combined with Gaussian priors for the tissue intensities, provides a model for the image to recover. Combined with a rigid registration (with Normal priors for translation and rotation) and an a-priori estimated point-spread function, the model is fitted to observed noisy data in a MAP sense using the ADAM optimizer.
Synthetic evaluation demonstrates clearly improved registration performance utilizing a Rician noise model over a Gaussian.
Evaluation on real Sodium MRI data shows no apparent improvement over: (i) no motion correction, and (ii) FLIRT based motion correction (Jenkinson et al, 2002).

Strengths:

The approach is straightforward, presumably novel (but see below), and the synthetic experiments provide good motivation for its usage.

Weaknesses:

- Reproducibility is substandard. Lacking information on the generated synthetic data (“from our generative model with known tissue parameters”). No information about the used point-spread function P. Lacking information on the segmentation (“These images were used for preparing tissue segmentation maps using SPM12 and co-registration with the uncorrected average Sodium image”). Also lacking information about the images (e.g. image sizes). Lacking information about the (ADAM) optimization (only LR and #iterations given).
Clarity is poor. The process is described in pieces and never presented as a whole. Symbols are introduced without explanation, or are explained later in the text.

- References to prior work can be improved. Please relate to more recent works. Please relate the work to [A] which is using a voxel-wise Rician ML estimation for motion correction. Also, more information about the comparison method (FLIRT) is needed.

- Language needs to be improved! There are several grammatical errors in the text, see examples below.

- Comparison with state of the art is very poor. The single method compared with is 20 years old. References to related methods are from 1997, 2001, 2002.

- Method performance on real data is unfortunately also poor. On 5 out of 8 samples, the method worsens the performance related to no motion correction at all. The overall judgement is that there is no apparent improvement from the used method.
However, the method of performance evaluation on real data, i.e., mean voxelwise standard deviation over the set of acquired images, may also be questioned, particularly in the light of the synthetic tests which show that a Gaussian noise prior is inappropriate (which typically relates to minimal standard deviation).


[A] G. Ramos-Llordén, A. J. den Dekker, G. Van Steenkiste, J. Van Audekerke, M. Verhoye and J. Sijbers, "Simultaneous motion correction and T1 estimation in quantitative T1 mapping: An ML restoration approach," 2015 IEEE International Conference on Image Processing (ICIP), 2015, pp. 3160-3164, doi: 10.1109/ICIP.2015.7351386.

---

### Official Review · Reviewer_u2yA · 2022-02-21

**Rating:** 5
**Confidence:** 3
**Recommendation:** Long Oral

**Deanonymize Review:**

no

**Detailed Comments:**

1. The proposed method depends on a segmentation map for which authors have acquired a high-resolution T1-weighted image. Is it always possible to acquire a high-resolution image of the same object? What is the advantage of acquiring low-resolution Sodium images?

2. Why is there not any prior on $\sigma$? Is it fair to assume that the $\sigma$ is the same for all images used for averaging? What all factors influence the noise? How do you initialize $\sigma$?

3. The mean of voxel-wise standard deviations is one way to understand the sharpness of the image. It could be informative to understand the distribution of pixel-intensities at a few pixels before and after registration. In the case of Rician noise, the mean of standard deviation may not be the best estimate to quantify the sharpness.

**Paper Type:**

methodological development

**Strengths Weaknesses:**

Strength:
1. Employing a generative modeling approach seems logical and sound for the problem considered in the paper.
2. The overall paper is well-written. There is a smooth flow from one section to another.
3. The experiments are well-thought. The results look promising.

Weakness:
1. The results with real MRI data show the resultant average image with the mean of voxel-wise mean deviation. Reporting other estimated parameters could also be beneficial for understanding the nature of the proposed method.

It will be appreciable if MRI data are made public for research purposes.

---

### Official Review · Reviewer_TbvJ · 2022-02-21

**Rating:** 2
**Confidence:** 5

**Deanonymize Review:**

no

**Detailed Comments:**

* What was the actual voxel size of the sodium images; 4mm^3 is mentioned, but are the voxels isotropic?
* It was not clear to me which data set helped generate the Gaussian priors over tissue intensities
* Have the authors considered evaluating tools other then FLIRT; eg; robust registration technique by M Reuter?

The submission proposes an interesting idea, but it needs to be flushed out with more details/ The presentation / introduction of
the Sodium images should be more thorough given that many performance characterization scores rely on these.

* for for image registration --> for image registration
* we can reduces --> we can reduce

**Paper Type:**

methodological development

**Strengths Weaknesses:**

* generative inference framework for motion correction
* interesting problem of very noisy image registration
* one requirement is the registration of a high resolution T1-weighted image to the one studied; how accurately can that registration be usually done
* Sodium imaging is not as well know as MRI / CT; some example images, representing average / typical image quality would have been very helpful; something similar to Fig 4 would have bee helpful
* The experimental section is somewhat limited
* What does it mean that the proposed tool works "well"? The Fig 3 plots display huge standard deviation and minimal difference between the original state, the newly proposed and the FLIRT solutions.
* The proposed tool makes some low motion examples worse. What is the reason for this?

---

### Official Review · Reviewer_wANt · 2022-02-22

**Rating:** 2
**Confidence:** 3

**Deanonymize Review:**

no

**Detailed Comments:**

No minor comments to provide.

**Paper Type:**

methodological development

**Strengths Weaknesses:**

This paper presents a simultaneous tissue identification and registration algorithm with the specific application of preforming motion correction in sodium imaging. For this, the authors derive an algorithm which is based on a Rician noise model instead of the usual Gaussian random noise. The authors develop a principled loss function to deal with this issue.

The main strengths are the justification of the problem as well as the formal derivation of the algorithm based on the Rician noise aspect of MRI. Nonetheless, there are several aspects which constitute important weaknesses for the paper

The main weakness of this paper is in the evaluation which is very succinct. For instance in figure 3 is quite hard to interpret the voxelwise std as a measure of performance, and when looking at the bar-charts the novel algorithm's improvement seems to be quite marginal with respect to the standard deviation. Furthermore the qualitative evaluation also seems marginal.

The simulated results however seem to provide good evidence of the potential of the approach. In consequence I fully recommend the authors provide a better MR-data based evaluation such that the advantages of their approach are brought into the light.

---

### Decision · Program_Chairs · 2022-02-22

**Decision:**

Accept

**Comment:**

While the majority of reviewers pointed out limited quantitative evaluation or demonstration of superiority on real data, they found merit in the method and positively commented on the interesting novel challenge that is being addressed. We therefore decided to accept the paper and expect that the authors will carefully improve the paper with respect to the comments on writing, discussion of related work and reproducibility (e.g. by releasing open source code).